# The Effect of Freeze–Thaw Cycles on the Microscopic Properties of Dumpling Wrappers

**DOI:** 10.3390/foods12183388

**Published:** 2023-09-10

**Authors:** Zhili Pan, Yibo Bai, Lina Xu, Yanjie Zhang, Mengmeng Lei, Zhongmin Huang

**Affiliations:** 1College of Food Science and Technology, Henan Agricultural University, Zhengzhou 450002, China; zp33@henau.edu.cn (Z.P.); byb220037@126.com (Y.B.); xln2008@126.com (L.X.); yanjiezhang@henau.edu.cn (Y.Z.); leimengmeng0418@163.com (M.L.); 2National R & D Center for Frozen Rice & Wheat Products Processing Technology, Zhengzhou 450002, China

**Keywords:** freeze–thaw, dumpling wrapper, water distribution, microstructure, gluten network

## Abstract

Dumplings are a traditional Chinese food welcomed by Chinese people. Research has indicated that process of quick-frozen wheat cultivars and their gliadins are all related to the quality and shelf-life of dumplings. Therefore, the effect of freeze–thaw cycles on the textural properties and microscopic characteristics of two types of quick-frozen dumpling wrappers (Zhaomai and Wenmai 19) and conformation of their gliadins were investigated. Scanning electron microscopy showed that Wenmai 19 dumpling wrappers had apparent damage after the first cycle, but Zhaomai wrappers did not reveal significant changes until the fourth cycle. The particle size distribution in the starch granules of Wenmai 19 wrappers varied in terms of mechanical damage, but Zhaomai delayed or avoided such effects. FT-IR found a loose protein structure of the gliadins. Differential scanning calorimetry showed that gliadins of Wenmai 19 degenerated more than those of Zhaomai. The crosslinking of gliadin and glutenin maintained a high-quality gluten network, thus protecting the gliadin stability from ice crystals. In turn, the gliadin maintained the strength of the gluten network. Therefore, raw flours with high-quality protein networks are more suitable for frozen dumplings. Freeze–thaw cycles dramatically decreased the textural characteristics of dumpling wrappers and the microscopic characteristics of their gliadin proteins. Concerning wheat cultivars with weak gluten, flours with high-quality protein networks are more suitable as raw materials for frozen dumplings.

## 1. Introduction

Dumplings are a traditional Chinese food with a long history, being welcomed for its flavor and balanced-nutrition in the accelerated pace of modern life. Dumplings are the largest frozen food category in China, accounting for one-third of the total frozen food market, and having had a swift growth rate. The market size of frozen dumplings in China in 2021 was about CNY 53.55 billion, an increase of 9.8% year-on-year, with a huge market share. Currently, the frozen dumpling market is shifting from low-end to high-end, and the food industry and consumers are demanding a higher quality of frozen dumplings.

As the main raw material, wheat flour determines the texture structure of dumpling wrappers. Wheat flour is mixed with water to form the dough, and the gluten protein contained in wheat flour, which is not found in other grains, forms the main network of the dough, endowing it with elastic, cohesive, and viscous properties [1,2]. Glutenin and gliadin become a three-dimensional protein network relying on secondary bonds and intermolecular disulfide bonds, giving the dough rich plasticity. Pastas require flours with gluten properties of different types; bread and dumplings need high-strength flour [3], while low-gluten wheats are suitable for making cakes and cookies. Gluten strength in flour is determined by the percentage of and ratio between the glutenin and gliadin [4].

Large numbers of studies have been conducted to relate the fractions of the gluten proteins. Generally, glutenin gives dough elasticity and hardness [5], and gliadin is sticky and has ductility [6]. Glutenin macropolymer (GMP), which is formed by glutenin agglomeration, affects the elasticity and strength of the dough. The higher the content, the larger the volume and the better the quality of the bread [7]. The molecular weight of gliadin, a monomeric protein, ranges from 30 to 80 kDa [8]. Barak found that additions of gliadin to dough resulted in softer dough, decreased stability, and increased cohesiveness [9]. Khatkar considered that when gliadin decreased the glass transition temperature of the dough, then the gluten network tended to be open [10]. A suitable gli/glu ratio was beneficial for bread-specific volume and hardness of bread crumbs, and the balance of gluten fractions was essential for enhancing bread quality, so gliadins are equally crucial as glutenins in assisting bread making [11]. The ratio of gliadin to glutenin has been demonstrated to impact the functional and rheological characteristics of wheat proteins. Upon hydration, gliadin transforms into a viscous liquid, contributing to the extensibility and viscosity of dough [12]. The lack of cysteine residues in ω-gliadins may affect the stability of the dough. These gliadin components may be suitable for making cookies with preferable texture and sensory aspects [13]. Liu demonstrated that only two gliadin genes, Gli-γ1-1D and Gli-γ2-1B, account for the majority of γ-gliadin accumulation. The author used knockout technology to improve end-use quality without affecting yield, thus resulting in valuable materials for biotechnology-based breeding programs [14]. Gliadin plays an extremely important role, but very little has been emphasized about the relationship between traditional pasta foods and gliadins, and the effect of gliadin on the microstructure of frozen dumplings has also not been reported.

Microstructure analysis is an ideal tool in the field of grain studies [15]. Scanning electron microscopy (SEM) analysis can intuitively provide graphics of protein, starch, and gluten networks. Weakening protein networks and splitting starch granules also offer effective identification and information about the impact of frozen storage and freeze–thaw cycles. The evolution of the number of dough pores and ice crystals can also be determined by software derivations [16]. In addition, particle size distribution, secondary structure analysis, and differential scanning calorimetry (DSC) also objectively prove the internal microstructure of dough and protein conformation.

China has made remarkable achievements in the breeding and production of wheat varieties in recent years [17], but the production of frozen dumplings is still in need of wheat flours of higher quality. Meanwhile, with the development of the economy and society, citizens’ eating habits have become more diversified, and multi-level requirements for pastas have occurred. Therefore, it is of practical significance to explore the relationship between wheat gliadins and dumpling wrappers. This study selected two kinds of wheat flours with high and low gluten to observe the changes in the microscopic properties of frozen dumplings and gliadins under freeze–thaw cycles to reveal the mechanism of quality deterioration.

## 2. Materials and Methods

### 2.1. Wheat Flour Samples

This study used two types of wheat cultivars (Zhaomai, Wenmai 19) with different gluten content from Henan Province, China. The two varieties of cleaned wheat were tempered with distilled water to a moisture content of 16% and left for 12 h at 4 °C. Then, the treated wheats were milled in a laboratory mill (LabMill, Chopin Technologies, Paris, France) to obtain flour. Zhaomai (16.62% protein, 47.79% wet gluten, 0.93% ash, 4.18% glutenin, 4.16% gliadin, 9.71% damaged starch) belongs to the high-gluten wheat type. Wenmai 19 (13.81% protein, 30.79% wet gluten, 0.36% ash, 2.51% glutenin, 2.26% gliadin, 10.83% damaged starch) belongs to the low-gluten wheat type. 

### 2.2. Preparation and Freeze–Thaw Cycles of Dumpling Wrappers

First, 40 g of distilled water and 1 g of salt were added to 100 g of wheat flour, and this was stirred at a constant rate for 10 min to form a dough. Then, we rested the dough in a sealed plastic bag at 25 °C for 30 min. Finally, the prepared dough was repeatedly folded and sheeted though an electric pasta machine (DMT-5, Longkou Fuxing Machinery Co., Ltd., Yantai, China) to a sheet with a dimension of length at 40 cm, width at 13 cm, and height at 1.5 mm. The sheets were cut into dumpling wrappers with a diameter of 6 cm, and then the wrappers were put through a freeze–thaw cycle process. The samples were temporarily stored in a desiccator after vacuum freeze-drying. Gliadins were extracted from some samples and temporarily stored in a desiccator after vacuum freeze-drying [18]. Furthermore, the vacuum freeze-drying conditions of samples were set as follows: temperature, −70 °C; vacuum degree 150 mTorr; freeze-drying time, 48 h.

### 2.3. Textural Properties Analysis

The dumpling wrappers’ hardness, springness, cohesiveness, and firmness were determined using a texture analyzer (SMS TA-XT2i, Stable Micro Systems Ltd, Surrey, UK). The TPA parameters were as follows: the probe was P/50; the pre-test speed was 2.0 mm/s; the test speed was 0.8 mm/s; the post-test speed was 0.8 mm/s; the compression ratio was 70%; and the trigger force was 5 g. The shear parameters were as follows: the probe was A/SFR; the pre-test speed was 2.0 mm/s; the test speed was 0.8 mm/s; the post-test speed was 0.8 mm/s; the compression ratio was 90%; and the trigger force was 5 g. Each measurement was repeated 10 times and averaged.

### 2.4. Particle Size Distribution

About 5 g of dumpling wrappers were ground for 2 min and put through a 120 mesh sieve, and then appropriate powder was added to the cuvette of Padmas laser particle size analyzer (Rise-2008; Jinan Runzhi Technology Co., Ltd, Jinan, China) using 95% ethanol as a dispersant. This was spiralled mixed for 1 min and ultrasonically vibrated for 1 min. After adjusting the baseline, test results were taken 7–10 times at random.

### 2.5. Water Distribution Analysis

Water distribution of quick-frozen dumpling wrappers were determined by LF-NMR (Low-Field Nuclear Magnetic Resonance, Shanghai Niumag Co., Ltd., Shanghai, China) according to the method described by Cao et al. [19] with some modifications. The sample (2.0 g) was sealed with parafilm and placed in a test tube with an inner diameter of 25 mm to avoid air and moisture interference. The determination parameters were as follows: CPMG (Carr-Purcell-Meiboom-Gill) pulse sequence, scanning frequency (SF) = 250 kHz, the number of sampling points (TD) = 100,104, time wait (TW) = 3000 ms, echo time (TE) = 0.3 ms, number of echoes (NECH) = 8000, pre-amp regulate gain (PRG) = 2, cumulative number = 8, 90° pulse width = 6 μs, 120° pulse width = 12 μs.

### 2.6. Scanning Electron Microscopy Observations (SEM)

Bulk samples were cut into cross-sections with a blade and treated with gold sputtering by the sputtering coating, and then they were placed into a scanning electron microscope (Hitachi S-3400N, Hitachi High-Tech Corporation, Tokyo, Japan) for observation and photographing at 1000 times (dumpling wrappers) and 5000 times (gliadins) magnification, respectively. The operating voltages were 5 kV and 10 kV.

### 2.7. Secondary Structure Study

About 5 mg of gliadin powder was added to 500 mg of dried potassium bromide (KBr) powder, then compressed by a tablet machine after sufficient grinding. The band scanning range was 400 cm^−1^ to 4000 cm^−1^, and the scanning was done 32 times on a Fourier transform infrared spectroscopy (FT-IR) (Nicolet iS50R, Thermo Fisher Scientific Inc., Waltham, MA, USA). Omnic 7.0 and Peak Fit v4.12 software analyzed the amide I bands (1600~1700 cm^−1^) in the infrared spectra. First, the baseline was corrected, and then the convolution was determined using the Gaussian method. Finally, multiple iterations derivations were progressed with r^2^ > 0.997; then, the contents of the secondary structure of the gliadin were calculated [20].

### 2.8. Thermal Properties Analysis

The thermal behavior of the gliadins was monitored by a DSC instrument(DSC 214, Netzsch Instrument Co. Ltd, Selb, Germany) with thermal Analysis Software [21]. The samples (3 mg) that mixed with distilled water (6 μL) were placed into NETZSCH pans and equilibrated for 24 h at 4 °C after being hermetically sealed. An empty sealed pan was used as a reference sample. The liquid N_2_ flow rate and the shielding gas flow rate were 20 mL/min. The temperature range used for the DSC analysis of dough samples was from 25 to 130 °C, and the heating rate was 10 °C/min. The freezable water of the quick-frozen dumpling wrapper was calculated from the ratio of the amount of frozen water to the total water in each sample. It can be expressed as Formula (1). The DSC profiles were treated with thermal Analysis Software.
(1)$$Freezable\;water\;(FW)=\frac{Hw}{Hi\times Tw}\times 100\%$$
where Hw (J/g) was the enthalpy of the samples; Hi was the known enthalpy of ice melting (334 J/g); and Tw (g/g) was the moisture content of the quick-frozen dumpling wrappers.

### 2.9. Statistical Analysis

The data were expressed as a mean ± standard deviation (SD). The data were analyzed (ANOVA and Duncan’s test) using Office 2007 (Microsoft, Redmond, WA, USA) and SPSS 20 software (IBM, Armonk, NY, USA). The probability value of *p* < 0.05 was considered significant.

## 3. Results and Discussion

### 3.1. Textural Properties of Dumpling Wrappers

Temperature fluctuations can significantly affect the quality of frozen pastas, as moisture migration and the growth of ice crystals can cause a range of mass drops [22]. Table 1 shows the hardness, springiness, cohesiveness, and firmness of the dumpling wrappers under freeze–thaw cycles, and there were significant differences between the two wheat cultivars. The hardness value of Zhaomai was much higher than that of Wenmai 19 under control conditions, because Zhaomai has a stronger network of protein. The hardness values of Wenmai 19 showed a continuous downward trend from the first to the fifth freeze–thaw cycles, with a clear decrease in the fifth freeze–thaw cycle, therefore proving that the process of the ice melting and regenerating considerably damaged the non-covalent bond and the gluten network; conversely, Zhaomai did not show a gradual downward trend until the fifth cycle.

The springness values of Zhaomai showed a similar trend to the hardness values throughout the freeze–thaw cycles, with a significant decrease (*p* < 0.05) by the fifth cycle, indicating that the damage of gluten was more serious. However, the springiness variations of Wenmai 19 were slight and irregular; this may indicate irregular changes in gluten conformation occurred. This also coincided with the results observed by SEM. Under the action of freeze–thaw cycle, the binding property of both wheat varieties showed a gradually decreasing trend, which was mainly associated with the degree of damage of the starch particles and gliadin [23]. Irregular and large ice crystals were harmful to the protein network and decreased the water retention capacity, and the expansion stress undermined the integrity of the starch granules. Meanwhile, the emancipation of amylase made the dumpling wrappers absorb excess water during cooking, causing an increased cohesiveness. A similar trend of firmness to the hardness of the frozen dumpling wrappers was shown under the freeze–thaw cycles.

### 3.2. Particle Size Distribution in Dumpling Wrappers

The samples were divided into different levels according to the percentage of the particle size distribution of each class. D_50_ was used to represent the average size of the particles, also called the median particle size. D_10_ and D_90_ were named in the same way. Previous studies have demonstrated that higher average particle sizes favor the spreading factor and decrease the dough hardness, and it is necessary to also have a certain percentage of fine particles, which being placed between coarser particles in the dough, give rise to a higher dough cohesiveness [24]. The particle size distribution of the dumpling wrappers under freeze–thaw cycles were as shown in Table 2. The D_50_ value of the two wheat varieties decreased significantly after 3–5 freeze–thaw cycles. The D_50_ value of Zhaomai showed a slight increase before the second freeze–thaw cycle, though this may have been due to the mechanical effects on protein molecules after the aggregation of the ice crystals. The D_50_ value of Wenmai 19 significantly decreased after the first freeze–thaw cycle (*p* < 0.05), then fell to the comparatively small value in the second freeze–thaw cycle and was unchanged afterwards, which may have been due to the ice crystals damaging the microstructure inside the wrappers [25], the weakening of the crosslinked gluten network, and the breaking of the integrity of the starch granules.

### 3.3. Water Distribution Analysis

As shown in Table 3, one freeze–thaw cycle had little effect on the relaxation time of T_21_ and T_22_, A_21_ and A_22_, but with the increase in the number of freeze–thaw cycles, the relaxation time and the peak areas exhibited significant differences for Zhaomai dumpling wrappers. The increase in relaxation time and A_22_ and the decrease in A_21_ indicated that the flow of water inside the dumpling wrappers was enhanced, with a decrease in the content of tightly bound water and an increase in that of softly bound water. In a study of Ding et al. [26], the destructive effects of freezing and freeze–thaw cycles were found to be higher on glutenin than on gliadin. These results may be related to the formation and redistribution of ice crystals during freeze–thaw cycles [27], resulting in the inhibition of ice re-crystallisation during freeze–thaw treatment. The relaxation time of Zhaomai samples was shortened, and the decline of A_21_ was even smaller compared with that of Wenmai 19 counterparts. The water that was tightly bound through hydrogen bonds inside the dumpling wrappers was less prone to be destroyed during the freeze–thaw cycles, thereby slowing down the deterioration rate in the quality of the quick-frozen Zhaomai dumpling wrappers.

### 3.4. Microscopic Analysis of Dumpling Wrappers

Figure 1 showed the dumpling wrappers’ SEM images (1000×) under freeze–thaw cycles. Under the control conditions, the microstructure of Zhaomai had a more developed gluten network and interwove with the starch granules to form a dense texture, while the protein network of Wenmai 19 was weak and more loosely connected to the starch granules. After the first freeze–thaw cycle, the internal structure of Wenmai 19 produced large holes, and much of the gluten had bubbly pores, which may have been due to an outflow of semi-free water from irregular ice crystals, weakening the gluten’s tight cross-linking [28], and leaving bubbly pores after it dried. By contrast, Zhaomai’s dense network inhibited the generation of large ice crystals better than Wenmai 19 did. After the third freeze–thaw cycle, the microscopic structure of Wenmai 19 showed destruction and deformation of a large number of starch granules, and at the same time, the protein network was divided into fragments [29]. Wu et al. [30] observed by CLSM that when the freeze–thaw storage reached a certain time, the gluten protein network structure appeared to be changed slightly, attributable to the loose of network structure and the emergence of uneven holes in the dough, which was basically consistent to what we observed through SEM. The most obvious transformations were shown in Zhaomai’s gluten, which produced numerous fragments in the fifth freeze–thaw cycle; the network structure was weakened, and more uneven holes in the dough were apparent, which showed that moisture migration occurred and that the internal structure was no longer stable. This result shows that the strength of ice crystal formation is much greater than the strength of the high-quality gluten network after multiple freeze–thaw cycles.

### 3.5. Microscopic Analysis of Gliadins

As shown in Figure 2, Zhaomai had a relatively dense structure, while Wenmai 19 contained many pores and was organized more loosely under control conditions. With the freeze–thaw cycles, Zhaomai primarily changed from dense to loose, gradually revealing a globular monomeric protein. Gliadin is a globular protein that forms intrachain disulfide bonds and acts as a filler in the gluten network [31]. Gliadin relies mainly on secondary bonds and a limited number of intramolecular disulfide bonds, which may be mechanical effects or freeze-concentration effects that untie gliadin’s secondary bonds, so they are part of the monomer protein. The microstructure of Wenmai 19 in the third cycle was better than the control treatment, probably because its hydrophobic group was exposed and the concentrated effect resulted in the agglomerated gliadin, thus bridging the loose structure. In the fifth freeze–thaw cycle, Wenmai 19 showed large numbers of monomeric proteins, proving that the gliadins were subject to more significant damage and widespread splitting of the secondary bonds, leading to dissociation of the polymer. Gliadin plays a role as filler and maintains the stability and strength of the gluten network with glutenin when it separates from glutenin due to the hydrophobic agglomeration and must make the gluten network open and weaken. The textural properties of the frozen dumpling wrappers also changed on the macroscopic level.

### 3.6. Secondary Structure of Gliadins

The secondary structure denotes the spatial structure of regular repeating conformation, including α-helix, β-turn, β-sheet, etc. At present, Fourier transform infrared (FT-IR) absorption spectroscopy was mainly used to analyze the secondary structure of the proteins. The secondary structure distribution of the gliadin under freeze–thaw cycles was as shown in Table 4. We found that the β-turn contents of Zhaomai rose under the freeze–thaw cycles, and the conformation of the gliadin molecular changed; thereby, the ordered structure of the gliadin turned into a disordered structure. The numbers of α-helix continued to decline, indicating that the freeze–thaw cycles had a harmful impact and that the gliadins lost some toughness [32]. The β-sheet contents of both Wenmai 19 and Zhaomai significantly decreased, implying that extension may occur in the gliadin [33]. The freeze–thaw cycles had a significant impact on the secondary structure of the gliadin, with a decrease in α-helical and an increase in β-turn; in addition, the molecule changed from a folded state to a stretch, with a transition from order to disorder, which indicated that the gliadin’s conformation profoundly changed. At the same time, the degree of crosslinking in the glutenin and gliadin decreased and somehow kept separate states [34], thus injuring the strength of the entire system of the frozen dumpling wrappers.

### 3.7. Thermal Properties of Gliadins

The denaturation temperature and enthalpy can reflect the degree of denaturation, hydrophobic interaction, and stability of proteins. Figure 3 shows the thermal properties of the gliadins under the freeze–thaw cycles. The denaturation temperature of Zhaomai tended to slow down. It explained how, under the freeze–thaw cycles, the protein natural conformation gradually developed into a disordered state and became weak in stability, which inevitably affected the protein network and pasta quality. These findings were consistent with FT-IR. The denaturation temperature of Wenmai 19 initially decreased and then rose rapidly, proving that its conformation had undergone a profound change. The ice stress gluten protein network dispersion and its internal hydrophobic groups were exposed to the surface, while the strong hydrophobic interactions promoted a rise in the denaturation temperature. The enthalpy of both Zhaomai and the Wenmai 19 went up, which described how the gliadin showed agglomeration and network stability as the freeze–thaw cycles gradually declined. With the increase in the times of freeze–thaw cycles, the content of freezable water in quick-frozen dumpling wrappers increased. After Wenmai 19 samples had gone through freeze–thaw cycles for more than three times, the content of freezable water was obviously increased. Ding et al. [35] reported that freeze–thaw treatment increased the content of freezable water in the dough, which could be attributed to ice recrystallisation during the freeze–thaw cycles. The gluten network with tighter crosslinking of gliadin and glutenin in the Zhaomai dumpling wrappers was less prone to be damaged by freeze–thaw cycles, leading to the lower freezable moisture content than Wenmai 19 samples.

## 4. Conclusions

The study showed that freeze–thaw cycles dramatically changed the textural characteristics of dumpling wrappers and the microscopic characteristics of their gliadin proteins. The initial ice crystals lowered the water-holding capacity of the system and enhanced the gliadin hydrophobic interactions. Then, as the more irregular ice crystals caused cleavage of the secondary bonds and changed the conformation, the molecular state became extensive, the degree of crosslinking between glutenin and gliadin dropped, the free starch granules were on the outside of the system, and the integrity was destroyed. The starch and protein components of the system formed a relatively independent state, which is damaging to the shelf life of frozen dumplings, so it is necessary to avoid circulation temperature fluctuations in the cold chain.

A developed gluten network structure can slow or delay deterioration in the quality of frozen dumplings during freeze–thaw cycles [36]. The textural properties of Zhaomai and Wenmai 19 flours were very different in that the texture curves of Zhaomai were flatter, while those of Wenmai 19 increased or decreased drastically. The nature of the protein network and gliadins are closely related to wheat flour pastas. The developed gluten network was able to maintain crosslinking between gliadins and glutenins well, thereby protecting the stability of gliadin. In turn, stable gliadin and glutenin kept the strength of the three-dimensional framework. Compared with wheat cultivars with weak gluten, flours with high-quality protein networks are more suitable as raw materials for frozen dumplings [37,38].

## Figures and Tables

**Figure 1 foods-12-03388-f001:**
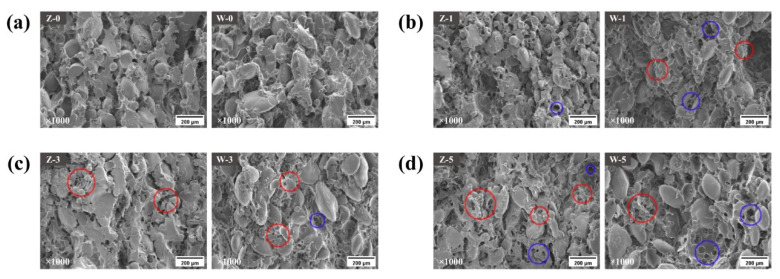
SEM images of dumpling wrappers in control treatment (**a**), first (**b**), third (**c**), and fifth (**d**) freeze–thaw cycle. Letters “Z” and “W” mean Zhaomai and Wenmai 19, respectively. Red circle, damaged protein network; blue circle, ice crystals.

**Figure 2 foods-12-03388-f002:**
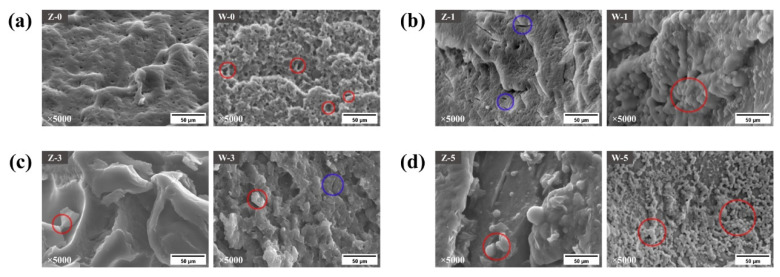
SEM images of gliadins in control treatment (**a**), first (**b**), third (**c**), and fifth (**d**) freeze–thaw cycle. Letters “Z” and “W” mean Zhaomai and Wenmai 19, respectively. Red circle, monomeric proteins; blue circle, damaged gliadin structure.

**Figure 3 foods-12-03388-f003:**
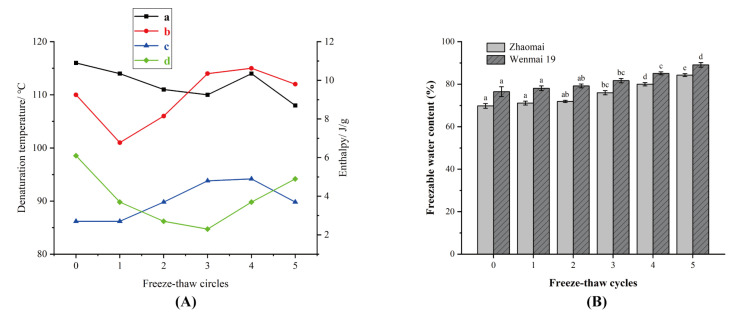
(**A**): Thermal properties of gliadins under freeze–thaw cycles. (a) Zhaomai’s denaturation temperature; (b) Wenmai 19’s denaturation temperature; (c) Zhaomai’s enthalpy; (d) Wenmai 19’s enthalpy. (**B**): Change of freezable water content of the quick-frozen dumpling wrappers during freezing and thawing; different lowercase letters show significant differences between Zhaomai and Wenmai 19 under the different number of freeze–thaw cycles (*p* < 0.05).

**Table 1 foods-12-03388-t001:** Textural properties of quick-frozen dumpling wrappers under freeze–thaw cycles.

	Freeze–Thaw Cycles	Hardness (g)	Springness	Cohesiveness	Firmness (g)
Zhaomai	0	15,277 ± 964 ^a^	0.992 ± 0.002 ^a^	0.843 ± 0.019 ^ab^	927 ± 154 ^a^
	1	14,811 ± 1124 ^a^	0.992 ± 0.002 ^a^	0.849 ± 0.007 ^a^	879 ± 3 ^ab^
	2	15,940 ± 1231 ^a^	0.990 ± 0 ^a^	0.837 ± 0.011 ^ab^	907 ± 24 ^a^
	3	16,105 ± 589 ^a^	0.993 ± 0.001 ^a^	0.824 ± 0.009 ^b^	767 ± 33 ^b^
	4	14,619 ± 220 ^a^	0.990 ± 0 ^a^	0.834 ± 0.005 ^ab^	875 ± 15 ^ab^
	5	11,505 ± 545 ^b^	0.973 ± 0.019 ^b^	0.849 ± 0.013 ^a^	804 ± 14 ^ab^
Wenmai 19	0	9000 ± 1165 ^a^	0.953 ± 0.009 ^a^	0.903 ± 0.006 ^a^	471 ± 48 ^ab^
	1	8404 ± 660 ^ab^	0.937 ± 0.044 ^a^	0.891 ± 0.005 ^a^	504 ± 81 ^a^
	2	7337 ± 589 ^bc^	0.948 ± 0.013 ^a^	0.877 ± 0.017 ^a^	439 ± 72 ^ab^
	3	6995 ± 406 ^cd^	0.889 ± 0.070 ^a^	0.845 ± 0.017 ^b^	403 ± 98 ^ab^
	4	7396 ± 785 ^bc^	0.957 ± 0.014 ^a^	0.881 ± 0.020 ^a^	403 ± 23 ^ab^
	5	5866 ± 295 ^d^	0.938 ± 0.043 ^a^	0.900 ± 0.031 ^a^	368 ± 5 ^b^

Note: The values labeled with different letters in the same column for the same sample showed significant differences (*p* < 0.05).

**Table 2 foods-12-03388-t002:** Particle size distribution of quick-frozen dumpling wrappers under freeze–thaw cycles.

	Freeze–Thaw Cycles	D_10_ (μm)	D_50_ (μm)	D_90_ (μm)
Zhaomai	0	10.48 ± 1.53 ^c^	23.27 ± 1.92 ^ab^	44.04 ± 0.62 ^a^
	1	13.86 ± 0.35 ^b^	26.19 ± 1.46 ^a^	40.86 ± 0.37 ^abc^
	2	16.36 ± 2.46 ^a^	25.03 ± 1.55 ^a^	35.24 ± 0.8 ^cd^
	3	8.91 ± 0.11 ^c^	20.49 ± 0.25 ^c^	33.12 ± 0.36 ^cd^
	4	10.65 ± 1.18 ^c^	20.48 ± 3.44 ^c^	36.52 ± 3.38 ^bcd^
	5	11.48 ± 1.2 ^bc^	20.46 ± 1.81 ^c^	42.16 ± 2.48 ^ab^
Wenmai 19	0	16.52 ± 0.08 ^a^	27.27 ± 0.76 ^a^	45.06 ± 2.63 ^a^
	1	13.35 ± 1.03 ^b^	20.58 ± 0.12 ^b^	34.33 ± 1.3 ^b^
	2	11.85 ± 0.87 ^bc^	18.92 ± 0.68 ^c^	32.91 ± 2.51 ^b^
	3	10.46 ± 1.46 ^cd^	18.51 ± 0.36 ^c^	29.07 ± 1.38 ^c^
	4	9.51 ± 0.37 ^d^	18.56 ± 0.36 ^c^	41.81 ± 1.61 ^a^
	5	11.63 ± 0.5 ^bc^	18.61 ± 0.62 ^c^	29.31 ± 1.69 ^c^

Note: The values labeled with different letters in the same column for the same sample showed significant differences (*p* < 0.05).

**Table 3 foods-12-03388-t003:** Water distribution and migration of quick-frozen dumpling wrappers under freeze–thaw cycles.

	Freeze–Thaw Cycles	T_21_ (ms)	T_22_ (ms)	A_21_	A_22_
Zhaomai	0	1.05 ± 0.07 ^c^	44.49 ± 0.01 ^c^	2716.17 ± 29.64 ^a^	15,888.18 ± 35.20 ^c^
	1	1.08 ± 0.03 ^c^	45.90 ± 0.04 ^c^	2607.52 ± 16.96 ^a^	16,094.73 ± 37.48 ^c^
	2	1.10 ± 0.02 ^c^	48.36 ± 0.01 ^c^	2553.20 ± 22.13 ^b^	16,364.83 ± 25.49 ^b^
	3	1.16 ± 0.01 ^b^	50.95 ± 0.02 ^b^	2444.55 ± 19.44 ^c^	16,682.59 ± 33.43 ^b^
	4	1.31 ± 0.04 ^a^	52.53 ± 0.04 ^b^	2254.42 ± 18.22 ^d^	17,318.12 ± 37.94 ^a^
	5	1.53 ± 0.03 ^a^	51.03 ± 0.01 ^a^	2145.77 ± 17.56 ^e^	17,635.88 ± 30.86 ^a^
Wenmai 19	0	1.12 ± 0.00 ^e^	54.79 ± 0.02 ^d^	2447.99 ± 14.50 ^a^	16,641.55 ± 37.02 ^c^
	1	1.14 ± 0.01 ^e^	56.57 ± 0.01 ^d^	2350.07 ± 18.73 ^a^	16,891.17 ± 29.51 ^c^
	2	1.19 ± 0.04 ^d^	59.84 ± 0.06 ^c^	2301.11 ± 25.62 ^b^	17,307.21 ± 32.82 ^b^
	3	1.25 ± 0.02 ^c^	62.40 ± 0.03 ^b^	2178.11 ± 26.66 ^c^	17,640.04 ± 26.87 ^b^
	4	1.43 ± 0.05 ^b^	64.33 ± 0.02 ^b^	2007.35 ± 17.46 ^d^	18,305.71 ± 35.26 ^a^
	5	1.72 ± 0.05 ^a^	70.13 ± 0.03 ^a^	1909.43 ± 19.71 ^e^	18,638.54 ± 25.39 ^a^

Note: The values labeled with different letters in the same column for the same sample showed significant differences (*p* < 0.05).

**Table 4 foods-12-03388-t004:** Secondary structure distribution of gliadins under freeze–thaw cycles.

	Freeze–Thaw Cycles	α-Helix (%)	β-Turn (%)	β-Sheet (%)
Zhaomai	0	24.12% ± 0.23% ^c^	22.21% ± 0.72% ^c^	15.90% ± 0.31% ^a^
	1	26.62% ± 0.22% ^a^	22.44% ± 0.55% ^c^	12.48% ± 0.66% ^d^
	2	26.17% ± 0.07% ^b^	23.69% ± 0.81% ^b^	12.95% ± 0.51% ^cd^
	3	26.03% ± 0.18% ^b^	24.68% ± 0.07% ^a^	13.70% ± 0.16% ^bc^
	4	26.04% ± 0.12% ^b^	23.35% ± 0.25% ^b^	13.34% ± 0.40% ^bc^
	5	26.18% ± 0.13% ^b^	25.01% ± 0.20% ^a^	13.91% ± 0.20% ^b^
Wenmai 19	0	28.00% ± 0.37% ^a^	26.08% ± 0.57% ^ab^	15.05% ± 0.47% ^a^
	1	26.72% ± 1.37% ^a^	23.73% ± 0.96% ^c^	14.79% ± 1.29% ^ab^
	2	27.63% ± 0.45% ^a^	26.04% ± 0.40% ^ab^	14.60% ± 0.66% ^ab^
	3	27.56% ± 0.46% ^a^	25.30% ± 0.24% ^b^	14.35% ± 1.05% ^ab^
	4	27.95% ± 0.21% ^a^	26.97% ± 0.46% ^a^	14.76% ± 0.11% ^ab^
	5	26.90% ± 0.29% ^a^	24.30% ± 0.20% ^c^	13.31% ± 0.74% ^b^

Note: The values labeled with different letters in the same column for the same sample showed significant differences (*p* < 0.05).

## Data Availability

The datasets generated for this study are available on request to the corresponding author.

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
