# Peer review of "The Effect of Freeze–Thaw Cycles on the Microscopic Properties of Dumpling Wrappers"

_foods, 2023, doi:10.3390/foods12183388_

Round 1

Reviewer 1 Report

Effect of Freeze-Thaw Cycles on Microscopic Properties of  Dumpling Wrappers and Gliadins

Zhili Pan, Yibo Bai, Lina Xu, Yanjie Zhang, Mengmeng Lei and Zhongmin Huang

 In the current study, the effect of freeze-thaw cycles on the textural properties and microscopic characteristics of two types of quick-frozen dumpling wrappers and the conformation of their gliadins were investigated.

Keywords

Please use other keywords which are not in the title.

Abstract

Lines 27 and 28: Do you mean glutenin instead of gluten since crosslinking occurs between gliadin and glutenin and gluten composed of glutenin and gliadin?

Introduction

Line 52 and 53 do you mean the ratio between glutenin and gliadin instead of gluten? Please be mindful of terminology throughout the manuscript.

Please give more details on how the gliadin component of gluten plays an important role in food quality.

Materials and Methods

Water mobility analysis is important for such a study. Please consider the determination of water mobility.

A confocal laser scanning microscope would be more informative as a microscopic analysis for the observation of the gluten microstructure.

 2.1. Wheat flour samples

Please consider the determination of gliadin, glutenin, and damaged starch contents of the flours.

 2.8. Statistical Analysis

Statistical analysis should have been done in a way to show the effects of both the protein content and freeze-thawing cycles. Please consider repeating the statistical analysis.

Results and Discussion

 Please elaborate on the discussion parts.

 3.3. Microscopic analysis of dumpling wrappers

Please explain the results in detail. Please show the networks, connections, and deformations on the SEM images to elaborate the explanation and deepen the discussion.

Lines 196 and 197 Please consider showing the fragments on the images and explaining them in detail.

 3.5. Thermal properties of gliadins

 Please give DSC thermo grams at least as a supplementary material.

Measurement of the freezable water content of frozen dough using differential scanning calorimetry would be helpful to elaborate the discussion.

 3.6. Microscopic analysis of gliadins

Please explain the results in detail. Please show the networks, connections, and deformations on the SEM images to elaborate the explanation.

Some sentences are difficult to understand because of the language. Thus, minor editing of the English language is required.

Reviewer 2 Report

Detail recomendations 

In introduction give more informations about effect on other factors technological on dumplings.

In materials and methods section give more informations about flour (chemical compositions ect.).

In my opinions you should made DSC analysis, color analysis and sensory analysis.

The disccussions should be improved. 
